# Pyrazole-Curcumin Suppresses Cardiomyocyte Hypertrophy by Disrupting the CDK9/CyclinT1 Complex

**DOI:** 10.3390/pharmaceutics14061269

**Published:** 2022-06-15

**Authors:** Masafumi Funamoto, Yoichi Sunagawa, Mai Gempei, Kana Shimizu, Yasufumi Katanasaka, Satoshi Shimizu, Toshihide Hamabe-Horiike, Giovanni Appendino, Alberto Minassi, Andreas Koeberle, Maki Komiyama, Kiyoshi Mori, Koji Hasegawa, Tatsuya Morimoto

**Affiliations:** 1Division of Molecular Medicine, School of Pharmaceutical Sciences, University of Shizuoka, 52-1 Yada, Suruga-ku, Shizuoka 422-8526, Japan; funamoto@u-shizuoka-ken.ac.jp (M.F.); y.sunagawa@u-shizuoka-ken.ac.jp (Y.S.); m-genpei@nms.ac.jp (M.G.); kana.smz.29@gmail.com (K.S.); katana@u-shizuoka-ken.ac.jp (Y.K.); s.shimi@jikei.ac.jp (S.S.); t.hamabe@u-shizuoka-ken.ac.jp (T.H.-H.); koj@kuhp.kyoto-u.ac.jp (K.H.); 2Division of Translational Research, Clinical Research Institute, National Hospital Organization Kyoto Medical Center, 1-1 Fukakusa Mukaihatacho, Fushimi-ku, Kyoto 612-8555, Japan; nikonikomakirin@yahoo.co.jp; 3Shizuoka General Hospital, 4-27-27-1 Kitaando, Aoi-ku, Shizuoka 420-8527, Japan; kiyoshimori2001@gmail.com; 4Dipartimento di Scienze del Farmaco, Università del Piemonte Orientale, Largo Donegani 2, 28100 Novara, Italy; giovanni.appendino@uniupo.it (G.A.); alberto.minassi@uniupo.it (A.M.); 5Michael Popp Institute and Center for Molecular Biosciences Innsbruck (CMBI), University of Innsbruck, 6020 Innsbruck, Austria; andreas.koeberle@uibk.ac.at; 6Graduate School of Public Health, Shizuoka Graduate University of Public Health, Shizuoka 420-0881, Japan; 7Department of Molecular and Clinical Pharmacology, School of Pharmaceutical Sciences, University of Shizuoka, Shizuoka 422-8526, Japan

**Keywords:** cardiomyocyte hypertrophy, p300, curcumin, histone acetyltransferase, Cdk9, phosphorylation

## Abstract

The intrinsic histone acetyltransferase (HAT), p300, has an important role in the development and progression of heart failure. Curcumin (CUR), a natural p300-specific HAT inhibitor, suppresses hypertrophic responses and prevents deterioration of left-ventricular systolic function in heart-failure models. However, few structure–activity relationship studies on cardiomyocyte hypertrophy using CUR have been conducted. To evaluate if prenylated pyrazolo curcumin (PPC) and curcumin pyrazole (PyrC) can suppress cardiomyocyte hypertrophy, cultured cardiomyocytes were treated with CUR, PPC, or PyrC and then stimulated with phenylephrine (PE). PE-induced cardiomyocyte hypertrophy was inhibited by PyrC but not PPC at a lower concentration than CUR. Western blotting showed that PyrC suppressed PE-induced histone acetylation. However, an in vitro HAT assay showed that PyrC did not directly inhibit p300-HAT activity. As Cdk9 phosphorylates both RNA polymerase II and p300 and increases p300-HAT activity, the effects of CUR and PyrC on the kinase activity of Cdk9 were examined. Phosphorylation of p300 by Cdk9 was suppressed by PyrC. Immunoprecipitation-WB showed that PyrC inhibits Cdk9 binding to CyclinT1 in cultured cardiomyocytes. PyrC may prevent cardiomyocyte hypertrophic responses by indirectly suppressing both p300-HAT activity and RNA polymerase II transcription elongation activity via inhibition of Cdk9 kinase activity.

## 1. Introduction

Heart failure is the most serious potential consequence of all cardiac diseases and is associated to high morbidity and mortality worldwide [1]. When the heart is subjected to external stresses, such as pressure or volume overload, cardiac hypertrophy occurs, which eventually can lead to heart failure [2]. Many studies have shown that cardiac hypertrophy is a strong predictor of heart failure [3,4]. Therefore, controlling cardiac hypertrophy is an important target for prevention and treatment of heart failure.

Although cardiomyocyte hypertrophy is a compensatory mechanism for stresses, such as hypertension and myocardial infarction (MI), continuous stress disrupts this mechanism and causes heart failure. Hypertrophic stress leads to cardiomyocyte hypertrophy through a variety of signaling pathways [5]. The transcriptional coactivator, p300, has histone acetyl transferase (HAT) activity and plays a major role in cardiomyocyte hypertrophy. The HAT activity of p300 is enhanced by hypertrophic stimulation, and p300 acetylates histones and transcription factors, such as GATA4, MEF2, and NFAT3, resulting in upregulation of atrial natriuretic factor (ANF) and brain natriuretic peptide (BNP) transcriptions [6,7,8]. MI surgery of transgenic (TG) mice exhibiting cardiac-specific p300 overexpression markedly enhanced left-ventricular (LV) remodeling after MI relative to that of wild-type (WT) mice. In TG mice with cardiac-specific overexpression of mutant p300 lacking HAT activity, LV remodeling after MI was attenuated to the same extent as in the WT mice [9]. Thus, p300-HAT activity could be a major target for heart-failure therapy.

The beneficial effects of natural products have been reported in many disorders, such as heart failure and cancer. Quercetin and butyric acid inhibit cancer cell growth [10,11], whereas curcumin and resveratrol suppress the development of heart failure [12,13]. In addition, several compounds and natural products, such as metformin, n-3 PUFA (DHA, EPA), anacardic acid, and curcumin (CUR), suppress cardiomyocyte hypertrophy by inhibiting p300-HAT activity [5,14,15,16]. CUR, a polyphenol derived from turmeric (Curcuma longa) in the ginger family, has a p300-specific inhibitory effect on HAT activity [17]. CUR possesses a variety of bioactivities, including anticancer, antioxidant, anti-inflammatory, and anti-Alzheimer’s effects [18]. The key pathways involving the beneficial effects of curcumin and its derivatives in cardiovascular diseases and cardiotoxicity mediated by anticancer drugs are reduced cardiac fibrosis and improved mitochondrial metabolism [19,20]. In addition, CUR inhibits acetylation of histones and GATA4 by inhibiting p300-HAT activity, thereby suppressing the transcriptional activity of cardiac hypertrophic response genes and cardiomyocyte hypertrophy [21]. In two rat models of heart failure with hypertension and MI, CUR ameliorated cardiac hypertrophy and dysfunction and suppressed development of heart failure by inhibiting p300-HAT activity [9,21]. These findings suggest that CUR may be a novel therapeutic agent for heart failure that targets p300-HAT activity. CUR, with many bioactivities, has attracted worldwide attention. In addition, many derivatives of CUR have been synthesized using CUR as a lead compound and are being actively investigated [22,23]. However, little research has been conducted on the structure–activity relationship of CUR using cardiomyocyte hypertrophy.

Curcumin pyrazole (PyrC) is a curcumin derivative where the α, β-unsaturated β-diketone element of CUR is replaced by a pyrazole ring, and prenylated pyrazolo curcumin (PPC) is its corresponding C-isoprenyl derivative (Figure 1). Pyrazole derivatives are relatively stable chemically and have pharmacological effects, such as antipyretic, analgesic, anticancer, and anti-inflammatory effects on humans, and they have been widely studied for their pharmaceutical potential [24]. On the other hand, the prenyl group can dramatically enhance the pharmacodynamics and the pharmacokinetic properties of polyphenols [25]. In fact, PyrC inhibits the inflammatory cytokine 12-lipoxygenase and the reporter activity of STAT3 more potently than CUR. PPC also has a more specific and very potent inhibitory effect on the inflammatory cytokine 15-lipoxygenase than CUR [24]. However, the effects of PyrC and PPC on cardiomyocyte hypertrophic response have not been investigated. We, therefore, compared the inhibitory effects of CUR, PyrC, and PPC on p300-HAT activity and cardiomyocyte hypertrophy, analyzing the inhibitory mechanism of cardiomyocyte hypertrophy.

## 2. Materials and Methods

### 2.1. Materials

CUR was purchased from Sigma-Aldrich (USA). PPC and PyrC were prepared as previously described [24]. Each of these compounds was dissolved in dimethyl sulfoxide.

### 2.2. Neonatal Rat Cardiomyocyte Culture

The study protocol was approved by the Animal Care and Use Committee of the University of Shizuoka (approved numbers: 176278). The 1–2-day-old Sprague Dawley rats used in the study were purchased from Japan SLC Inc. (Shizuoka, Japan). Primary cardiomyocytes were isolated from neonatal Sprague-Dawley rats as described previously [26]. Briefly, isolated cardiomyocytes were cultured in Dulbecco’s Modified Eagle Medium (DMEM; L1N7624) (Sigma-Aldrich, St. Louis, MO, USA) containing 10% fetal bovine serum (Sigma-Aldrich) and stabilized penicillin/streptomycin solution (Nacalai Tesque, Kyoto, Japan) at 37 °C for 48 h. CUR, PPC, or PyrC were added to cardiomyocytes. After 2 h, 30 μM phenylephrine (PE) (Fuji-film Wako Pure Chemical Corporation, Osaka, Japan) was added and the cultures were incubated for an additional 48 h to induce cardiomyocyte hypertrophy.

### 2.3. Immunofluorescence Staining and Measurement of the Surface Area of Cardiomyocytes

Immunofluorescence staining was performed with antimyosin heavy-chain antibody (PA0493, Leica Biosystems, Wetzlar, Germany) and Alexa555-conjugated antimouse immunoglobulin (Ig)G antibody (# A28180, Invitrogen, Waltham, MA, USA) as previously described [5]. In brief, cardiomyocytes were fixed with 3.7% formaldehyde for 15 min, blocked in 1% bovine serum albumin and 0.5% nonyl phenoxypolyethoxylethanol-40/Tris-buffered saline-Ca for 1 h, then incubated with antimyosin heavy-chain antibody overnight. The cells were then incubated with Alexa555-conjugated antimouse IgG for 2 h and stained with Hoechst33432 (# H341, Nacalai Tesque, Kyoto, Japan) for 1 h. Immunofluorescence was observed using a LSM 510 laser scanning microscope (Carl Zeiss, Jena, Germany). The cell surface areas of 50 randomly chosen cells in each group were measured using ImageJ software v4.16 (Bethesda, MD, USA).

### 2.4. qRT-PCR

Total RNA from cultured cardiomyocytes was extracted using the TRI reagent (Nacalai Tesque, Japan) and reverse transcribed into cDNA using ReverTra Ace^®^ qPCR RT Master Mix (Toyobo, Osaka, Japan). Quantitative PCR was performed using the KOD SYBR qPCR mix (Toyobo, Japan) with an ABI 7500 Real-Time PCR System (Applied Biosystems, Waltham, MA, USA). The following primers were used:
rat-ANF Fw  ATCACCAAGGGCTTCTTCCTrat-ANF Rv  CCTCATCTTCTACCGGCATCrat-BNP Fw  TTCCGGATCCAGGAGAGACTTrat-BNP Rv  CCTAAAACAACCTCAGCCCGTrat-18S Fw  CTTAGAGGGACAAGGGGGrat-18S Rv  GGACATCTAAGGGCATCACA

### 2.5. Western Blotting

Whole-cell extracts, nuclear extracts, histone extracts, and a sample from the in vitro HAT assay were prepared and Western blots were performed as described previously [26]. For Western blotting, rabbit polyclonal anti-acetyl-histone H3K9 antibody (#9649, Cell Signaling Technology, Danvers, MA, USA), rabbit polyclonal antihistone H3 antibody (#9715, Cell Signaling Technology, USA), rabbit polyclonal anti-RNA Polymerase II antibody (sc-47701, Santa Cruz, Dallas, TX, USA), goat polyclonal anti-Cdk9 antibody (C-20) (sc-484, Santa Cruz, USA), rabbit polyclonal anti-Cyclin T1 antibody (sc-10750, Santa Cruz, USA), mouse monoclonal anti-β-actin clone AC-15 IgG (A1978, Sigma-Aldrich, USA), goat antirabbit IgG–HRP antibody (MBL, Tokyo, Japan), and sheep antimouse IgG (GE Healthcare, Chicago, IL, USA) were used. The signals were detected by a C-DiGit Chemiluminescent Western Blot Scanner (LI-COR, Lincoln, NE, USA) and a LAS-1000 Plus luminescent image analyzer (Fujifilm, Tokyo, Japan) and quantified using Image Studio LITE software (LI-COR, USA).

### 2.6. In Vitro HAT Assay

The in vitro p300-HAT assay was performed as described previously [5]. In brief, purified p300-HAT recombinant protein and 5 μg of core histones from calf thymus (Worthington, Lakewood, CA, USA) were incubated in HAT buffer (50 mM Tris-HCl (pH 8.0), 10% glycerol, 0.1 mM EDTA (pH 8.0), 1 mM DTT), in the presence of CUR, PPC, or PyrC at room temperature for 30 min. Acetyl-CoA was added to each sample and incubated for 30 min. All samples were subjected to Western blotting.

### 2.7. Glutathione S-Transferase (GST) Pull-Down Assay

Recombinant proteins were prepared using pGEX-Cdk9 and pDEST17-Cyclin T1, and GST pull-down assay was performed as previously described [27,28]. Briefly, GST-Cdk9 was incubated with glutathione-Sepharose 4B beads (GE Healthcare, USA). The beads containing equal amounts of GST or GST-Cdk9 were mixed with His6-Cyclin T1 in wash buffer (20 mM Tris-HCl, 10% glycerol, 0.1% Tween 20, 100 mM KCl, 5 mM MgCl_2_, 10 mM 2-mercaptoethanol, 0.2 mM EDTA, and 0.25 mM phenylmethylsulfonyl fluoride, pH 8.0). The mixtures were incubated with low-speed rotation at 4 °C for 2 h. The beads were washed four times with wash buffer. The bound material was eluted with sample buffer (0.1 M Tris-HCl, 4% sodium dodecyl sulfate (SDS), 20% glycerol, 0.002% bromophenol blue, 12% 2-mercaptoethanol), and separated by SDS-polyacrylamide gel electrophoresis (PAGE). GST fusion proteins were visualized with Coomassie brilliant blue staining.

### 2.8. Statistical Analysis

The results are expressed as the means ± standard error. Statistical comparisons between experimental groups were performed using two-way ANOVA followed by the Tukey test. The results were considered significant if *p* was <0.05.
**Figure**  **F Statistics**Figure 2B  134.649Figure 2C  14.561Figure 2D  23.910Figure 2F  10.950Figure 3B  37.009Figure 4C  21.750Figure 5C  909.750Figure 5F  254.105Figure 5H  215.074Figure 6C  14.245

## 3. Results

### 3.1. PyrC Suppressed PE-Induced Cardiomyocyte Hypertrophy at Lower Concentrations Than CUR

To investigate whether PPC and PyrC could suppress PE-induced cardiomyocyte hypertrophy, primary cultured cardiomyocytes were used. As shown in Figure 2A,B, PyrC at 1 and 3 μM significantly inhibited cardiomyocyte hypertrophy, which was observed with CUR at 3 and 10 μM. Additionally, the level of suppression of cardiomyocyte hypertrophy was comparable between 10 μM PPC and 10 μM CUR. PE-induced increase in the mRNA levels of ANF and BNP was significantly suppressed by 3 μM PyrC to the same extent as by 10 µM CUR, whereas 10 µM PPC did not significantly suppress these mRNA levels (Figure 2C,D). Moreover, 3 μM PyrC significantly suppressed the acetylation levels of H3K9 by PE to the same extent as 10 µM CUR. On the other hand, 10 μM PPC did not significantly suppress these acetylation levels (Figure 2E,F). These results suggest that PyrC suppressed PE-induced cardiomyocyte hypertrophy at a lower dose than CUR.

### 3.2. PyrC Inhibited the Phosphorylation of RNA Polymerase II in Cardiomyocyte Hypertrophy

To determine if PyrC can directly inhibit p300-HAT activity, an in vitro p300-HAT assay was performed. The results showed that 60 μM CUR significantly inhibited H3K9ac, whereas 200 μM PyrC and PPC did not significantly inhibit it (Figure 3A,B). These results indicated that PyrC did not directly inhibit p300-HAT activity in vitro.

### 3.3. PyrC Did Not Suppress H3K9ac In Vitro

Kinases regulate p300-HAT activity by several upstream signals, such as Akt, extracellular signal-regulated kinase (ERK) 1/2, p38 mitogen-activated protein kinase, and cyclin-dependent kinase 9 (Cdk9) [27,28,29,30]. To identify a mechanism for suppressing p300-HAT activity by PyrC, we focused on various kinases in p300 upstream signals. First, we checked the effect on the phosphorylation of Akt, ERK1/2, and p38 by CUR, PPC, and PyrC in cultured cardiomyocytes. The results showed that CUR, PPC, and PyrC did not suppress the PE-induced phosphorylation of Akt, ERK1/2, and p38 (Figure 4A). Next, to evaluate the kinase activity of Cdk9, we investigated the phosphorylation of RNA polymerase II as a substrate phosphorylated by Cdk9. Phosphorylation levels of RNA polymerase II were suppressed to the same extent by 10 µM CUR and 3 μM PyrC (Figure 4B,C). These results showed that PyrC regulated the kinase activity of Cdk9 in cultured cardiomyocytes.

### 3.4. PyrC Inhibited the Binding of Cdk9 with Cyclin T1

Activation of Cdk9 is regulated by the interaction between Cdk9 and Cyclin T1 [31]. Therefore, to assess if 3 μM PyrC and 10 μM CUR could affect the formation of the Cdk9/Cyclin T1 complex in cultured cardiomyocytes. PyrC and CUR suppressed the interaction of Cdk9 with Cyclin T1 (Figure 5A–C). Formation of the Cdk9/Cyclin T1 complex was enhanced by PE, whereas CUR and PyrC suppressed the PE-induced increased interaction between Cdk9 and Cyclin T1 (Figure 5D–F). In addition, to determine if PyrC could directly inhibit the interaction of Cdk9 with Cyclin T1, we performed a GST pull-down assay with recombinant 6 X His-Cyclin T1 and GST-Cdk9. As shown in Figure 5G,H, PyrC and CUR directly inhibited the interaction of Cdk9 with Cyclin T1. Moreover, PyrC suppressed the interaction of Cdk9 with Cyclin T1 at lower concentrations than CUR. These results indicated that PyrC regulated p300-HAT activity in cultured cardiomyocytes by disrupting the Cdk9/Cyclin T1 complex.

### 3.5. PyrC Suppressed Phosphorylation of p300

To determine if PyrC could suppress phosphorylation of p300, a target of Cdk9 in cardiomyocytes, immunoprecipitated proteins-WB was performed. PyrC suppressed phosphorylation of p300 at lower concentrations than CUR (Figure 6A–C). These results indicated that PyrC inhibited the kinase activity of Cdk9 in cultured cardiomyocytes.

## 4. Discussion

This study shows that PyrC inhibits cardiomyocyte hypertrophy at a lower concentration than CUR, and that PPC lacks this activity. PyrC could suppress histone acetylation during PE-induced cardiomyocyte hypertrophy but could not directly inhibit p300-HAT activity in vitro. However, PyrC inhibited the kinase activity of Cdk9 by disrupting the Cdk9/Cyclin T1 complex. These results suggest that PyrC indirectly inhibits p300-HAT activity by inhibiting Cdk9, the upstream regulator of p300.

In this study, we focused on Cdk9 as a target of PyrC, since PyrC did not directly inhibit p300-HAT in an in vitro HAT assay. Cdk9 forms a complex with Cyclin T1 and functions as a protein kinase complex called a positive transcription elongation factor b p-TEFb [31]. This complex enhances transcription elongation by phosphorylating the second serine residue of the repeat sequence (Y-S-P-T-S-P-S) in the C-terminal domain of RNA polymerase II [32]. Cdk9 alone is highly unstable, exists stably, and is activated by forming the Cdk9/Cyclin T1 complex [33], which suggests that formation of a complex with other factors is important for Cdk9 activity [34]. Furthermore, Cdk9 is seemingly involved in cardiomyocyte hypertrophy, since Cdk9 kinase activity is essential for PE- or ET-1-induced cardiomyocyte hypertrophic response [28,35]. Cdk9 binds directly to p300, phosphorylates p300, and upregulates p300-HAT activity upon cardiac hypertrophic stimulation 21. Continuous activation of the Cdk9/Cyclin T1 complex induces not only cardiomyocyte hypertrophy but also cardiac apoptosis by disrupting mitochondrial function, suggesting that the Cdk9/Cyclin T1 complex is a novel target for heart-failure therapy [36,37]. This study showed that PyrC regulates Cdk9 activity more potently than CUR by suppressing formation of the Cdk9/Cyclin T1 complex and that PyrC inhibits cardiomyocyte hypertrophy by inhibiting not only p300-HAT activity but also RNA polymerase II phosphorylation. Thus, PyrC is potentially beneficial for heart-failure therapy, although further studies are necessary to validate these findings clinically.

Previous studies have shown that p300-HAT activity has a critical role in the development of cardiac remodeling during heart failure and is a major risk factor for heart failure [12,26]. Akt, located downstream of PI3K, enhances p300-HAT activity by phosphorylating the 1834 serine residue of p300 [27]. SIRT 6, a histone deacetylase of the sirtuin family, promotes proteasome-dependent degradation of p300 by suppressing PI3K/Akt signaling, and negatively regulates the expression level of p300. Moreover, cardiac hypertrophy is reportedly observed in SIRT6-knockout mice [38,39]. ERK1/2, an important signaling pathway in the pathogenesis of heart failure, stabilizes p300 and increases its expression by phosphorylating serine residues of p300 [29]. In cardiomyocytes, p38 is overexpressed and promotes the degradation of p300 during the development of heart failure [30]. In addition, SIRT1, which is highly expressed in the heart, promotes SUMOylation of p300 by de-acetylating p300, and is involved in the degradation of p300 [13]. These data suggest that p300 is regulated by various binding proteins, but the possibility that PyrC regulates proteins upstream of p300 other than Cdk9 cannot be ruled out. Future studies should focus on other p300-binding proteins to clarify the mechanism of indirect inhibition of p300-HAT activity by PyrC.

PyrC could inhibit cardiomyocyte hypertrophy at a lower concentration than CUR by suppressing cdk9. On the other hand, PPC did not inhibit cardiomyocyte hypertrophy. This is seemingly associated to differences in the structure of these compounds. CUR suppress p300-HAT activity and reportedly inhibits cardiomyocyte hypertrophy and the development of heart failure [23]. In addition, there have been extensive structure–activity studies on the natural product CUR, and studies targeting p300-HAT activity and cardiomyocyte hypertrophy have been carried out [9]. It has been suggested that the role of the α, β-unsaturated β-diketone moiety of CUR is important for inhibition of p300-HAT activity [5], and both PyrC and PPC, devoid of this moiety, are indeed devoid of inhibitory activity on p300-HAT. On the other hand, this change enhanced the anti-inflammatory effect of PyrC more than of CUR. Specifically, PyrC suppressed LPS-induced inflammatory responses in macrophages by inhibiting JNK activation more potently than CUR [25]. PyrC also exhibited anti-inflammatory effects by inhibiting LPS-induced nuclear translocation of NF-κB and phosphorylation of p38 in microglial cells [40].

Curcumin has low solubility and low oral absorption; therefore, improving its absorption is an important issue in efforts for its effectiveness in in vivo utilization [41]. Ingestion of 8 g of natural curcumin powder increased the peak serum concentration to 1.77 ± 1.87 μM, showing a biological effect of curcumin in chemoprevention without toxicity. Thus, ingestion of large doses of curcumin is necessary to obtain a clinical effect [42]. Various efforts to overcome this issue have aimed to improve curcumin absorption, such as PLGA-based drug delivery, lipid nanoemulsions, coating with polysaccharides, and amorphous solid dispersion [43,44,45,46,47,48,49]. In addition, various studies have been aimed to develop curcumin derivatives with improved absorption in cells and in vivo and to improve efficacy in diseases [5,25]. Further examinations are needed to design curcumin formulations with high absorption efficiency and develop/formulate curcumin derivatives with improved tissue specificity and efficacy against diseases.

## 5. Conclusions

In this study, it was found that structural changes may strengthen the effect of PyrC on Cdk9. Taken together, our results validate CUR as a lead compound for the development of therapeutic agents for heart failure.

## Figures and Tables

**Figure 1 pharmaceutics-14-01269-f001:**
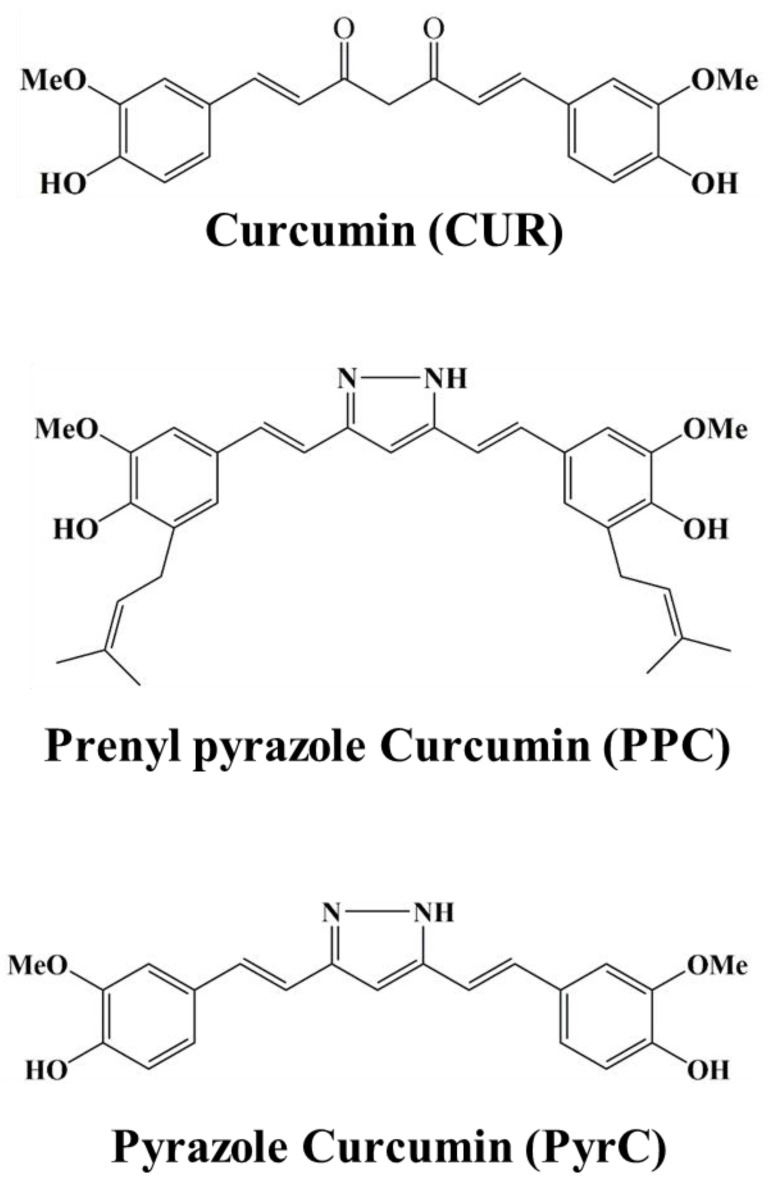
The chemical structure of curcumin and its analogs.

**Figure 2 pharmaceutics-14-01269-f002:**
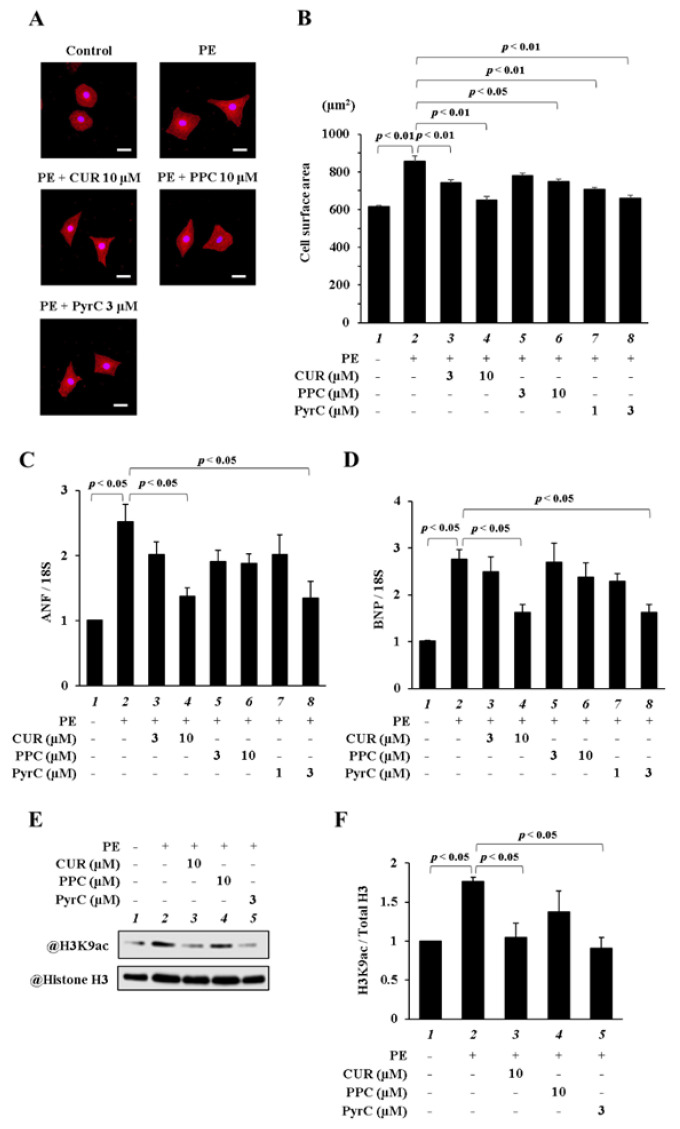
PyrC suppressed cardiomyocyte hypertrophy. (**A**,**B**) MHC immunostaining of primary cardiomyocytes. (**B**) Quantification of (**A**). Scale bar, 20 μm; N = 3. (**C**,**D**) mRNA levels of ANF (**C**) and BNP (**D**). N = 3. (**E**,**F**) Western blotting showing acetylation levels in cardiomyocytes. (**F**) Quantification of (**E**). N = 3.

**Figure 3 pharmaceutics-14-01269-f003:**
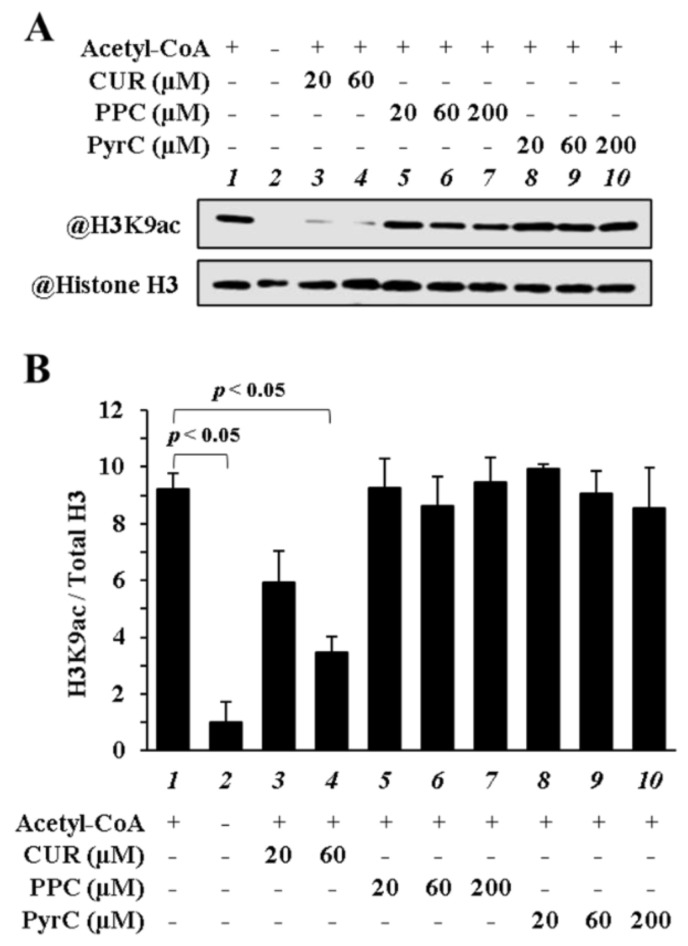
PyrC did not inhibit p300-HAT activity in vitro. (**A**,**B**) In vitro p300-HAT assay was performed using recombinant p300-HAT, histones, CUR (20, 60 µM), PPC (20, 60, 200 µM), and PyrC (20, 60, 200 µM). These acetylation levels of histone were examined by Western blotting. (**A**) Photograph of a representative Western blot. (**B**) Quantifications of (**A**). N = 3.

**Figure 4 pharmaceutics-14-01269-f004:**
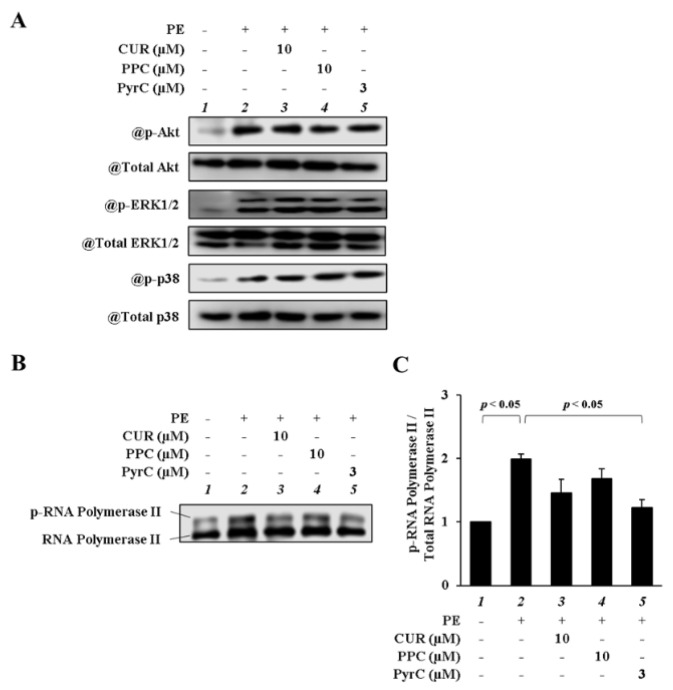
PyrC suppressed the phosphorylation of RNA polymerase II in cardiomyocytes. Primary cultured cardiomyocytes were treated with CUR (10 µM), PPC (10 µM), or PyrC (3 µM) and stimulated with 30 μM of PE. Whole-cell fraction (15 min after PE stimulation) and nuclear fraction (24 h after PE stimulation) were extracted from theses cardiomyocytes, and all samples were subjected to Western blotting. (**A**) Representative Western blot for p-Akt, total Akt, p-ERK, ERK, p-p38, and total p38. (**B**) Representative Western blot for p-RNA polymerase II and RNA polymerase II. (**C**) Quantifications of (**B**). N = 3.

**Figure 5 pharmaceutics-14-01269-f005:**
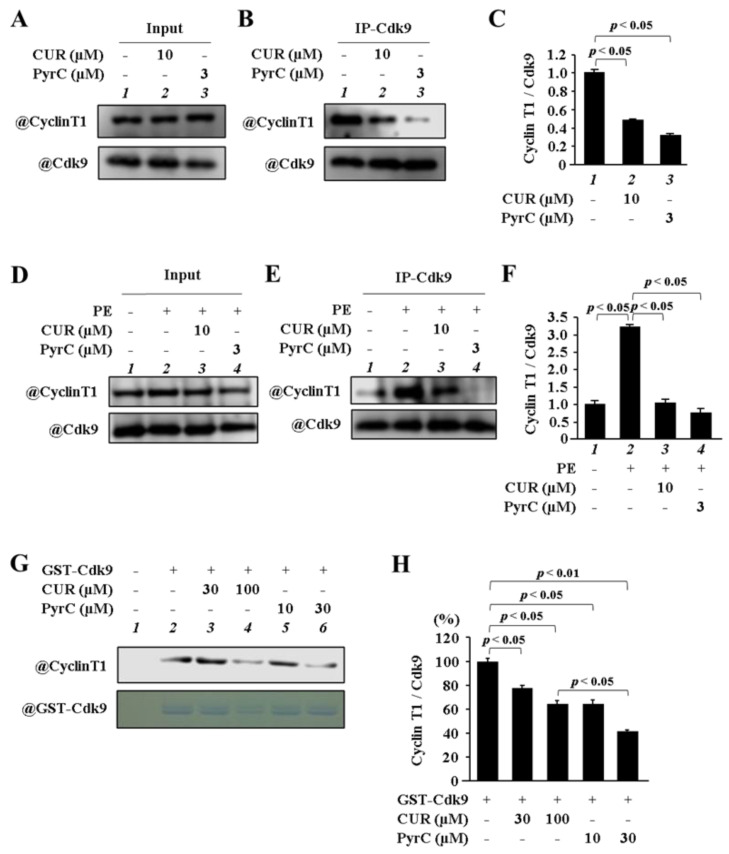
PyrC and CUR reduced the formation of Cyclin T1/Cdk9. (**A**–**C**) Primary cultured cardiomyocytes were treated with CUR (10 µM) or PyrC (3 µM). (**B**) Photograph of a representative IP-WB. (**C**) Quantifications of (**B**). N = 3. (**D**,**E**) Primary cultured cardiomyocytes were treated with CUR (10 µM) or PyrC (3 µM) and stimulated with 30 μM of PE. (**E**) Photograph of a representative IP-WB. (**F**) Quantifications of (**E**). N = 3. (**G**,**H**) GST pull-down assay was performed using recombinant 6XHis-cyclin T1 and GST-cdk9. (**G**) Photograph of a representative IP-WB. (**H**) Quantifications of (**G**). N = 3.

**Figure 6 pharmaceutics-14-01269-f006:**
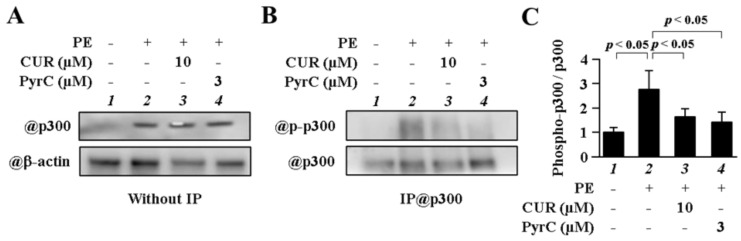
Phosphorylation of p300 was inhibited by PyrC in cardiomyocytes. (**A**–**C**) Nuclear proteins were extracted from primary cardiomyocytes after 48 h of PE stimulation. (**B**) Photograph of a representative IP-WB. (**C**) Quantifications of (**B**). N = 3.

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
