# Peer review of "Pyrazole-Curcumin Suppresses Cardiomyocyte Hypertrophy by Disrupting the CDK9/CyclinT1 Complex"

_pharmaceutics, 2022, doi:10.3390/pharmaceutics14061269_

Round 1

Reviewer 1 Report

The manuscript is of merit and interest since it provides a valuable and extensive work on the protective role of curcumin and its derivatives against hypertrophy. The authors employ a vast variety of experimental approaches. I congratulate the authors for the work proposal, but the following changes should be achieve previously to the ms. publication

Line 24: “To evaluate if prenylated pyrazolo curcumin (PPC) and 23 curcumin pyrazole (PyrC) can suppress cardiomyocyte hypertrophy. Cultured cardiomyocytes 24 were treated with CUR, PPC, or PyrC and then stimulated with phenylephrine (PE)” Remove the full stop between hypertrophy and cultured to make sense

Line 31: inhibits instead of inhibited

Line 71: for its pharmaceutical potential (include its)

Line 102: Include Ref. Number of DMEM to let the reader know the exact composition. Don´t you supplement the media with antibiotics?

Line 105: Indicate the reason of using Phenilephrine stimulation (to induce cardiomyocyte hyperthrophy)

Line 117: incubation time with Hoechst

Indicate the ref. number of antibodies used for immunofluorescence and western blot

Line 195: Include the value of the scale bar after Photograph of representative cardiomyocytes (in brackets)

Which charge control has been used to quantify the western blot results?

Author Response

Please see the attachment (Reviewer 1).

Reviewer 2 Report

Manuscript titled "Pyrazole-curcumin Suppresses Cardiomyocyte Hypertrophy by Disrupting the CDK9/CyclinT1 Complex" is a very interesting research article describing the key role of pyrazole-curcumin complex in cardioprotection. The mansucript have a good overall structure, methods are clear, results are clear and of good quality. References are updated but still to be improved in some parts. Authors should improve the manuscript in some parts:

1) Improve the introduction with better description of the beneficial effects of nutraceuticals like curcumin,  resveratrol, quercetin, boswellic acid and small chain fatty acids like butyric acid in cardiomyocytes and also against cancer cell growth ( cite 10.1002/jcp.25283 and 10.3892/or.2018.6932)

2) a better description of the key pathways of curcumin in cardiovascular diseases and cardiotoxicity mediated by anticancer drugs ( i.e reduction of ferroptosis and improvement of mithocondrial metabolism)

3) a better description,in discussion, of the main pharmacokinetic limitations of curcuminoids and the role of nanoemulsions to improve oral bioavailability of curcuminoids ( like PLGA-based drug delivery, lipid nanoemulsions coated with chitosan and others) ( cite 10.1016/j.nano.2016.08.022)

Author Response

Please see the attachment (Reviewer 2).

Reviewer 3 Report

In this article, the authors attempt to show that curcumin, and derivatives of curcumin, are beneficial in an in vitro model of cardiac hypertrophy.

Unfortunately, the paper appears to be a little premature as several critically important parts of the methods are absent.

1) Was cur and derivatives present during PE treatment?

2) DMSO was used as the vehicle and drugs were then diluted in aqueous media. Cur is not soluble in aqueous media - what is the solubility of the cur derivatives used here in DMSO and in aqueous media?

3) The authors state that 50 cells were analysed for size. As there appear to be 8 groups, this means approximately 6 cells per group were analysed, possibly 18 in total (N=3), which is unacceptable.

4) 1-way ANOVAs have been used and are clearly inappropriate for the type of outcome measures discussed in most figures, which includes treatment or no treatment, 3 separate but related drugs, and at least two levels of concentration within each drug.

5) The doses of drugs appear to change depending upon the outcome measure, for example, concentrations of up to 200uM are discussed for Figure 3, without explanation. Even within a particular figure (e.g., Fig 5), several different concentrations of cur are used, with no apparent explanation (compare H versus F versus C).

6) The high concentrations used for the experiments are not relevant for in vivo as curcumin is heavily metabolised (administration is typically oral) and reaches much lower concentrations in plasma (low nM) than were used here. Why were lower concentrations, which are far more biologically relevant, not used?

7) N=3 for the majority of experiments - is this reflected in the graphs? Thus, are the means+variation shown in graphs the mean+variation of 3?

Author Response

Please see the attachment (Reviewer 3).

Round 2

Reviewer 3 Report

Unfortunately, I find that the authors have not addressed all of my comments.

N=3 for the majority of experiments - is this reflected in the graphs? Thus, are the means+variation shown in graphs the mean+variation of 3? Are the graphs a mean of technical replicates or of actual biologic replicates?

F statistics are not provided for ANOVAs. Please provide them.

The authors' explanation for the large variance in dose is unsatisfactory. Figure 5 is reliant upon preceding data but the data from Figure 5G and H do not use the same concentrations as in previous graphs; this is a key piece of data and forms the title of the paper.
For example curcumin inhibits the proteasome at these extremely high concentrations, if it were to be used in vivo or in cells.

These results have questionable relevance to the clinic. The authors' comments on in vivo dosing of curcumin is unsatisfactory. The authors suggested a paper to help support their dosing, however, this paper found that serum peaks were approximately 2uM after ingesting 8g of curcumin (patients refused higher amounts).
Thus, when using unadulterated curcumin, these doses used here appear to be unfeasible for the clinic.

The authors final (new) sentence states: "to develop and formulate curcumin derivatives with improved tissue specificity and efficacy against diseases." The paper does not appear to provide evidence on improved tissue specificity. Moreover, the effects of curcumin and curcuminoids on e.g., the proteasome, at micromolar levels, must be appropriately considered.

Round 3

Reviewer 3 Report

I thank the authors for their responses.

I would ask that the authors place, in the manuscript, the F statistics.

I would ask that the authors place, in the manuscript, their discussion on dosing of curcumin, and relevance to the clinic.
